# Palliative Performance Scale Predicts Survival in Patients with Bone Metastasis Undergoing Radiotherapy

**DOI:** 10.3390/cancers18010061

**Published:** 2025-12-24

**Authors:** Gina Hennig, Emma Thrandorf, Dirk Vordermark, Jörg Andreas Müller

**Affiliations:** Department of Radiation Oncology, University Hospital Halle (Saale), D-06120 Halle (Saale), Germany; gina.hennig@student.uni-halle.de (G.H.); emma.thrandorf@student.uni-halle.de (E.T.); dirk.vordermark@uk-halle.de (D.V.)

**Keywords:** bone metastasis, palliative radiotherapy, palliative performance scale, survival

## Abstract

Accurately estimating survival is crucial for guiding treatment decisions in patients receiving palliative radiotherapy. In this retrospective real-world study, we evaluated the prognostic value of the Palliative Performance Scale (PPS) in 153 patients treated with radiotherapy for bone metastases. We found that functional status, as measured by PPS, was strongly associated with overall survival, with patients showing a PPS of 60% or higher living significantly longer. Completion of the planned radiotherapy course was also a key predictor of improved survival, even in a frail population. In contrast, PPS did not predict discharge destination after treatment, suggesting that post-treatment care needs are influenced by additional social and organizational factors. These findings support the routine use of PPS to aid prognostication and treatment planning in palliative radiotherapy.

## 1. Introduction

The global burden of cancer is steadily increasing, and with it the need for effective palliative care is becoming ever more urgent [1]. However, the adoption of palliative services shows substantial regional disparities: while some countries have successfully integrated palliative care into their healthcare systems, more than half of the world’s population still lacks access, and oncology-specific palliative structures remain scarce [2]. In this context, radiotherapy (RT) plays a central role as a palliative treatment modality, aiming not only at local tumor control but also at symptom relief and improved quality of life in patients with advanced cancer [3].

A key challenge in palliative oncology is accurate prognostication, particularly in end-of-life care. Reliable survival prediction tools are crucial for individualized treatment planning, shared decision-making, and allocation of healthcare resources. To address this need, Anderson and colleagues introduced the Palliative Performance Scale (PPS) in 1996 as a modification of the Karnofsky Performance Status (KPS), providing a structured measure of physical functioning and a potential tool for prognostication in palliative care settings [4].

Since its introduction, the PPS has been extensively studied and validated. Psychometric validation demonstrated its reliability and clinical utility in prognostic assessment, care planning, and interdisciplinary communication [5]. Moreover, the PPS has been widely translated and psychometrically validated across multiple languages and cultural contexts, demonstrating its robustness and international applicability in diverse healthcare systems [6,7,8,9].

Beyond validation, the PPS has also been shown to be interchangeable with other widely used performance status scales. Comparative studies have demonstrated strong correlations between PPS, KPS, and Eastern Cooperative Oncology Group (ECOG) scales, allowing flexible integration of PPS within established prognostic indices such as the Palliative Prognostic Score (PaP) and the Palliative Prognostic Index (PPI) [10]. Importantly, several investigations have confirmed the utility of PPS for survival prediction across heterogeneous hospice populations and disease entities, supporting its role as a robust prognostic tool in palliative care [11].

Despite this evidence, the application of PPS in heterogeneous cohorts of patients receiving palliative RT has not yet been systematically studied. Given that RT constitutes one of the most important modalities in symptom control and life-prolongation in advanced oncological disease [12], the integration of PPS as a prognostic tool in this setting is of particular interest. This retrospective analysis therefore investigates the prognostic value of the PPS in cancer patients undergoing palliative RT, with a focus on survival prediction and clinical utility in routine care.

## 2. Methods

### 2.1. Data and Material

This retrospective, single-center study was conducted at the Department of Radiation Oncology, University Hospital Halle (Saale), Germany. Patient data were collected from the digital archives of the Department of Radiation Oncology for the period January 2021 to April 2025. All patients who received palliative-intent RT for bone metastases during this period were eligible for inclusion, regardless of the underlying primary tumor type.

Clinical data were extracted from the hospital information system ORBIS (version 03.20.02.01), while RT parameters were obtained from Elekta Mosaiq (version 2.84). Imaging data were reviewed using Centricity PACS (GE Healthcare). Data were fully anonymized prior to analysis. The study was conducted in accordance with the ethical standards of the institutional and/or national research committee and with the 1964 Declaration of Helsinki and its later amendments. The study was approved by the Ethics Committee of the Medical Faculty of Martin Luther University Halle-Wittenberg (reference number: 2025-201).

Demographic and clinical characteristics included age at the start of RT, sex, and marital status (single, married, divorced, widowed, unknown). Functional status was assessed using the PPS. The PPS was assessed at the start of RT by a physician as part of the standard clinical evaluation. The PPS is a validated tool that quantifies functional status and prognostic performance in palliative care patients [4]. It ranges from 0% (death) to 100% (full health) in 10% increments, evaluating five domains: ambulation, activity level and evidence of disease, self-care, oral intake, and level of consciousness. Lower PPS scores indicate poorer functional capacity and are associated with shorter survival. The PPS was used to describe baseline performance status and as a covariate in the statistical analyses. In cases where the proportional hazards assumption was violated, the PPS was applied as a stratification variable in the final Cox regression model.

Overall comorbidity burden was assessed using the Charlson Comorbidity Index (CCI) and the age-adjusted CCI. The CCI is a well-established prognostic tool that estimates patient survival based on the number and severity of comorbid conditions. The age-adjusted version incorporates age-related weighting to improve prognostic accuracy in older patients [13]. An overview of the CCI scoring system, including the comorbidities considered and their respective weights in the index, is provided in Appendix A. The Body Mass Index (BMI) was calculated as weight (kg) divided by height squared (m^2^) and classified according to WHO categories: underweight (<18.5 kg/m^2^), normal weight (18.5–24.9 kg/m^2^), and overweight/obese (≥25 kg/m^2^) [14].

The cancer type was grouped into major oncologic categories, including gastrointestinal, lung, genitourinary, breast, gynecological, head and neck, hematologic, bone/soft tissue, and melanoma. RT data included the fractionation scheme (12 × 3 Gy, 10 × 3 Gy, 5 × 4 Gy, or 10 × 2 Gy) and treatment completion status (complete vs. incomplete). RT was considered completed when the full prescribed course was delivered. Systemic therapy prior to or concurrent with RT was classified into chemotherapy, endocrine therapy, immunotherapy, targeted therapy, or bone-directed therapy.

The discharge destination following RT was categorized as home with specialized outpatient palliative care, inpatient palliative care, rehabilitation, acute care/inpatient hospital, death, or other/unknown.

### 2.2. Statistical Analyses

The primary endpoint was overall survival (OS), defined as the time from the end of RT to death. Patients alive at the last documented contact were censored. The secondary endpoint was the discharge destination, analyzed dichotomously as home vs. other.

For survival analysis, Kaplan–Meier estimates were generated, and median OS with 95% confidence intervals (CI) was reported for the total cohort and relevant subgroups.

In addition, the restricted mean survival time (RMST) was calculated to provide an absolute measure of average survival time over the observed follow-up period. RMST was included because it does not rely on the proportional hazards assumption and offers a clinically interpretable summary of survival, particularly relevant in palliative cohorts with short and heterogeneous survival times and limited long-term follow-up.

Univariable and multivariable Cox proportional hazards regression models were used to identify prognostic factors associated with OS. Covariates included sex, age, marital status, PPS, BMI category, CCI, and RT completion. The proportional hazards assumption was verified using Schoenfeld residuals. Because the assumption was violated for the PPS, the final multivariable Cox model was stratified by PPS (<60 vs. ≥60) to allow for separate baseline hazards across performance status groups. Hazard ratios (HR) and corresponding 95% confidence intervals (CI) were estimated. Model discrimination was evaluated using the concordance statistic (C-index).

A logistic regression model was performed to identify predictors of discharge to home versus other destinations. Odds ratios (OR) with 95% CIs were reported for both univariable and multivariable analyses.

All statistical analyses were performed using RStudio (version 2024.04.2+764). A *p*-value < 0.05 was considered statistically significant. During the preparation of this work, the authors used ChatGPT (GPT-4, OpenAI, San Francisco, CA, USA) to assist in improving the clarity and language of the manuscript. All content was reviewed and edited by the authors, who take full responsibility for the final version.

## 3. Results

### 3.1. Patient Characteristics

A total of *n* = 153 patients who received palliative RT for bone metastases between 2021 and 2025 were included in the analysis. The median age at the start of RT was 67 years (±13), and 63% of patients were male. More than half of the cohort were married (58%), while 12% were widowed and 9.8% divorced.

Baseline functional status, assessed using the PPS, showed that most patients had moderate to good performance status: 52% had a PPS of 70%, while 32% scored 50%. Only 10% of patients had a PPS ≤ 40%.

The median BMI was 24.45 kg/m^2^ (IQR 5.15), with 49% of patients classified as having normal weight, 43% as overweight or obese, and 7.9% as underweight. The median CCI was 9 (range 6–16), reflecting a high comorbidity burden in this palliative population.

The most frequent primary cancer sites were genitourinary (28%), lung (23%), breast (17%), and gastrointestinal (12%) malignancies. RT fractionation schemes were predominantly hypofractionated, with 12 × 3 Gy (46%) and 10 × 3 Gy (34%) being the most common regimens. RT was completed as planned in 91% of patients.

Regarding systemic therapy, 42% of patients received chemotherapy, 14% immunotherapy, 11% targeted therapy, and 5% bone-directed therapy, while 25% had no systemic treatment documented.

At discharge, 35% of patients returned home with specialized outpatient palliative care, 14% were transferred to inpatient palliative care, 5% to rehabilitation, and 2% to acute care or inpatient hospital settings. Three patients (2%) died during treatment, and 41% had other or undocumented discharge destinations.

Detailed baseline characteristics of the study cohort are summarized in Table 1.

### 3.2. Survival Analyses

In the overall cohort, the median overall survival (OS) was 108 days (3.6 months) with a 95% confidence interval (CI) of 78–143 days (2.6–4.7 months). Given the palliative treatment intent and the absence of structured follow-up after completion of RT, the median follow-up time corresponded to the median OS. The restricted mean survival time was 233 days (7.7 months). All 126 patients had died during the observation period.

In univariable Cox regression analyses, no significant associations with OS were observed for age, marital status, BMI category, or comorbidity burden (CCI). However, male sex and RT completion showed trends toward significance. In the multivariable Cox model, male sex was associated with a higher risk of death (HR 1.61, 95% CI 1.06–2.46, *p* = 0.027), and higher age was linked to shorter survival (HR 0.98, 95% CI 0.96–1.00, *p* = 0.050). Patients with a PPS of 60% or higher had a significantly lower risk of death compared to those with PPS below 60% (HR 0.62, 95% CI 0.41–0.93, *p* = 0.021).

The strongest association was found for RT completion, which was independently associated with improved survival (HR 0.06, 95% CI 0.03–0.12, *p* < 0.001). No significant associations were identified for BMI category or CCI in the adjusted model.

Detailed results of the univariate and multivariable Cox proportional hazards analyses are presented in Table 2.

Kaplan–Meier survival analysis demonstrated a trend toward longer OS for patients with a PPS of 70% or higher compared to those with PPS below 70%. However, the difference between the two groups was not statistically significant (*p* = 0.18), as reflected by overlapping confidence intervals. The survival curves are presented in Figure 1. Supplementary Kaplan–Meier curves illustrating OS stratified by BMI categories (Appendix A), age groups (Appendix A), and sex (Appendix A) are provided.

In addition, overall survival events stratified by cancer subtype are summarized in Table 3 highlighting substantial heterogeneity in event rates across tumor entities, with particularly high event proportions observed in gastrointestinal, head and neck, and gynecological malignancies.

### 3.3. Logistic Regression Discharge Destination

In the logistic regression analysis assessing factors associated with discharge to home, none of the examined variables showed a statistically significant association. In the multivariable model, neither sex, age, marital status, PPS, nor BMI category was independently related to the likelihood of being discharged home. Age showed a non-significant trend toward higher odds of home discharge with increasing age (OR 1.03, 95% CI 1.00–1.06, *p* = 0.11). All other covariates, including marital status, PPS, and BMI, demonstrated odds ratios close to unity, indicating no meaningful effect. Detailed results of the univariate and multivariable logistic regression analyses are presented in Table 4.

### 3.4. Sensitivity Analysis

In the sensitivity analysis, the proportional hazards assumption was evaluated using Schoenfeld residuals. The initial Cox model showed a violation of this assumption for the PPS (*p* < 0.001). Therefore, PPS was included as a stratification factor in the final multivariable Cox regression model. After stratification, no significant violations of the proportional hazards assumption were detected for any variable (global test: *p* = 0.55), confirming adequate model fit.

In the final stratified model, male sex remained significantly associated with a higher risk of death (HR 1.86, 95% CI 1.20–2.88, *p* = 0.006), while increasing age was independently linked to shorter survival (HR 0.98, 95% CI 0.96–1.00, *p* = 0.015). Completion of the prescribed RT course was a strong independent predictor of improved survival (HR 0.04, 95% CI 0.02–0.10, *p* < 0.001). Marital status, BMI category, and CCI were not significantly associated with OS in the adjusted model.

Detailed results of the stratified Cox proportional hazards regression model are presented in Table 5.

## 4. Discussion

This retrospective study examined prognostic factors for OS in 153 patients who received palliative RT for bone metastases. The median OS of 3.6 months reflects the advanced disease stage typical of patients referred for palliative RT. Despite a high comorbidity burden, treatment completion was achieved in 91% of cases, emphasizing that palliative RT remains feasible and well tolerated even in a frail population. Among the investigated parameters, performance status (PPS) and RT completion were the strongest predictors of survival, while age and male sex were associated with poorer outcomes.

Our findings confirm the prognostic relevance of functional performance, as measured by PPS, in patients with advanced cancer. Numerous studies have demonstrated a strong correlation between PPS (or related scales such as Karnofsky or ECOG) and survival across different care settings. In the large cohort analyzed by Allende-Pérez et al., both Karnofsky and ECOG performance status were significantly associated with OS, underscoring the value of functional assessment for timely palliative referral and end-of-life (EOL) decision-making [15]. Similarly, Baik et al. summarized in their systematic review that PPS consistently predicts survival across heterogeneous palliative populations, with clearly graded reductions in life expectancy for declining PPS levels [16].

Prospective analyses by Cai et al. and Downing et al. confirmed this relationship, showing that higher PPS scores correspond to significantly longer survival, independent of diagnosis or setting [17,18]. This prognostic gradient was also confirmed in Asian populations: Prompantakorn et al. and Vankun et al. demonstrated similar stepwise survival reductions with decreasing PPS in Thai cohorts, supporting the scale’s validity across cultural and healthcare contexts [19,20].

Emerging evidence suggests that monitoring PPS trajectories provides more accurate prognostic information than a single baseline measurement. In particular, an early decline within the first two weeks has been shown to predict significantly poorer survival, regardless of initial PPS [21,22,23]. Medeiros et al. similarly reported that serial PPS assessments capture functional decline and help identify patients entering the terminal phase earlier than single-timepoint measures [24]. These dynamic trends underline the importance of repeated performance assessments during palliative RT, where rapid changes in patient condition are common.

Campos et al. demonstrated high inter-rater reliability of PPS in outpatient palliative RT settings, supporting its clinical usability among multidisciplinary teams [25]. Dewhurst et al. further showed how PPS correlates with frailty measures such as the Clinical Frailty Scale, allowing better communication across specialties [26]. Together, these findings confirm PPS as a reliable, standardized, and easily interpretable instrument for assessing prognosis and functional decline.

The prognostic power of PPS has been compared with composite scores and clinician judgment. Hui et al. found that PPS performed comparably to the PPI, PaP, and clinician prediction of survival, with similar accuracy for short-term survival estimates [27]. This supports the continued clinical relevance of PPS due to its simplicity and widespread familiarity. Moreover, PPS-based triggers for key care discussions, as suggested by Myers et al., may facilitate earlier advance care planning and alignment of therapeutic goals [28].

Educational interventions have also been shown to improve the use of PPS in clinical routine: Fedel et al. reported that teaching nurses how to apply PPS enhanced their confidence and accuracy in identifying patients in need of palliative care, highlighting its role not only in prognostication but also in team communication [29].

Our results align with previous findings demonstrating that performance status, rather than comorbidity burden or BMI, is the most meaningful survival predictor in advanced cancer populations. Leiros et al. showed that lower PPS values correlate with poorer nutritional markers, reflecting a multidimensional decline that may not be captured by anthropometric measures alone. This supports our observation that BMI categories did not significantly influence survival [30].

Consistent with Downing et al. [18], we found that male sex was independently associated with shorter OS, a pattern also observed by Lee et al. [23] in Korean hospice patients. Age likewise demonstrated a negative association with survival, mirroring results from several large-scale studies of performance-based prognosis [15,20].

The strong association between RT completion and improved survival in our cohort aligns with the findings of Yamaguchi et al., who demonstrated that patients with poor baseline performance (ECOG 3–4) can still benefit symptomatically and prognostically from palliative RT if treatment is completed [12]. Together, these results underscore that even in patients with reduced functional capacity, well-tolerated and goal-oriented RT can contribute to maintaining quality of life and possibly prolonging survival.

The prognostic accuracy of PPS highlights its potential role not only for outcome prediction but also for guiding clinical pathways and timing of palliative interventions. Kang et al. showed in a large randomized trial that early, structured palliative care improved quality of life and coping skills and was associated with a trend toward prolonged survival [31]. Together with our findings, these results support routine PPS-based assessments to identify patients who may benefit from earlier integration of comprehensive palliative services.

Several prognostic models have been developed to estimate life expectancy in patients referred for palliative radiotherapy, most notably the Survival Prediction Score (SPS) by Chow et al., the TEACHH model by Krishnan and Epstein-Peterson et al., and the NEAT model by Zucker and Tsai et al. [32,33,34]. Despite differences in complexity and included variables, all models consistently identify performance status as a central determinant of survival.

Chow et al. demonstrated that a simplified model incorporating primary tumor site, metastatic pattern, and Karnofsky Performance Status (KPS), with a threshold of KPS ≤ 60, effectively stratifies patients into prognostic groups [32]. This closely aligns with the PPS cutoff applied in the present study, supporting the clinical relevance of this functional threshold. Similarly, the TEACHH and NEAT models confirmed performance status as one of the strongest predictors of survival, alongside additional clinical and laboratory parameters [33].

In contrast to these composite scores, the present study emphasizes a pragmatic approach using PPS as a single, routinely assessed measure within a radiation oncology setting. Our findings suggest that PPS captures essential prognostic information comparable to more complex models, while offering superior feasibility in daily clinical practice. This supports the use of PPS as an efficient and clinically meaningful tool for survival estimation and treatment planning in patients undergoing palliative RT.

In contrast to its strong prognostic value for overall survival, PPS was not significantly associated with discharge destination in our cohort. Neither PPS nor other baseline clinical variables predicted discharge to home versus other destinations. This suggests that discharge disposition after palliative radiotherapy is influenced by factors beyond baseline functional and clinical status, including social support, caregiver availability, organizational aspects of post-acute care, and rapid clinical changes following treatment. The high proportion of patients discharged to other or unspecified destinations (41%) may have further limited the ability to detect significant associations. Overall, these findings indicate that PPS alone is insufficient to anticipate discharge needs, highlighting the importance of incorporating social and dynamic clinical factors in future studies.

A major strength of this study is its real-world, methodologically rigorous design, reflecting an unselected palliative population treated within an RT department. The systematic and standardized assessment of the PPS allowed for the integration of an established prognostic tool into clinical RT practice. By combining functional performance data with treatment variables such as RT completion, this study provides novel and clinically actionable insights into survival determinants in patients receiving palliative RT.

To our knowledge, this is the first study to evaluate the prognostic value of the PPS in a retrospective RT cohort, thereby bridging prognostic research in general palliative care with the specific context of palliative radiation oncology. This approach extends the evidence base for PPS beyond hospice and home-based care, demonstrating its feasibility, prognostic validity, and clinical utility in an oncologic treatment setting.

The strong association between completion of RT and improved survival, however, must be interpreted with caution. It is likely that this relationship does not solely reflect a causal treatment effect, but rather a selection bias: patients who were able to complete their prescribed RT course probably had a more favorable baseline condition and prognosis. Conversely, those who discontinued treatment prematurely often did so because of rapid clinical deterioration or imminent death. Thus, incomplete RT may act as a surrogate marker for poor performance status and limited life expectancy rather than an independent determinant of outcome.

Another limitation concerns the heterogeneous composition of the cohort, both in terms of PPS distribution (not restricted to patients in an end-of-life situation) and primary diagnoses, which included malignancies with widely differing biology, treatments, and prognoses. While this heterogeneity limits comparability with disease-specific studies, it also represents a clinical strength, as it mirrors the diversity of patients encountered in routine palliative RT. The finding that PPS retained prognostic significance despite this variability supports its robustness and general applicability across tumor entities and care contexts.

Further limitations include the retrospective, single-center design, which restricts external validity, and the lack of additional prognostic variables such as symptom burden, nutritional status, or psychosocial factors. Although PPS is a validated and reliable instrument, potential inter-rater variability cannot be excluded. Finally, survival analyses were limited to the available follow-up period, precluding long-term outcome assessment.

## 5. Conclusions

This study demonstrates that the PPS is a clinically meaningful and easily applicable prognostic tool in patients receiving palliative RT for bone metastases. A PPS score of ≥60% was associated with significantly longer survival, highlighting its value for individualized treatment planning and shared decision-making in palliative oncology. Among other variables, completion of the prescribed RT course emerged as the strongest independent predictor of improved survival, emphasizing the importance of treatment continuity even in advanced disease stages. In contrast, comorbidity burden, BMI, and marital status showed no significant association with survival outcomes.

By integrating PPS into routine clinical assessment, clinicians can more accurately estimate prognosis, facilitate appropriate care transitions, and optimize the balance between therapeutic benefit and treatment burden in patients with limited life expectancy. Future prospective studies should further validate PPS-based prognostic models across different tumor entities and palliative treatment settings.

## Figures and Tables

**Figure 1 cancers-18-00061-f001:**
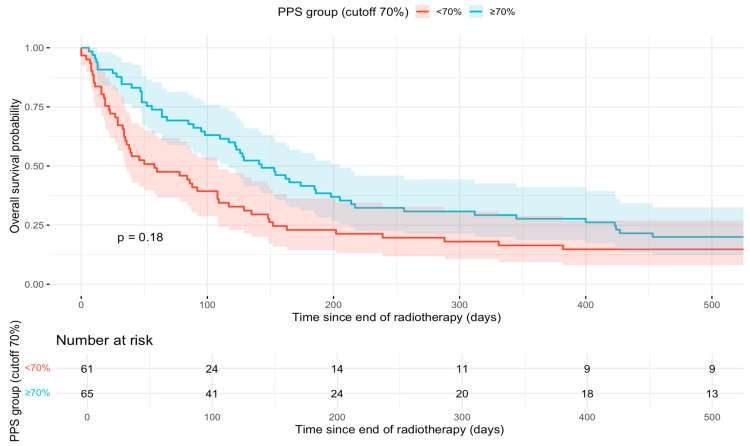
Kaplan–Meier survival curves according to Palliative Performance Scale (PPS) group. Kaplan–Meier estimates of overall survival (OS) are shown stratified by PPS group (<60% vs. ≥60%) at the end of radiotherapy.

**Table 1 cancers-18-00061-t001:** Baseline characteristics of patients included in the dataset. Values are presented as *n* (%) for categorical variables and as median (range) for continuous variables, as appropriate. Abbreviations: BMI, Body Mass Index; CCI, Charlson Comorbidity Index; PPS, Palliative Performance Scale; RT, Radiotherapy; The variable “Body Mass Index (BMI) Category” was defined according to WHO criteria (Underweight < 18.5 kg/m^2^; Normal weight 18.5–24.9 kg/m^2^; Overweight/Obese ≥ 25 kg/m^2^). The “Palliative Performance Scale (PPS)” was categorized into clinically meaningful groups (<40, 40–60, ≥70). The “Charlson Comorbidity Index (CCI)” is shown as median (range).

Variable	N = 153
**Sex, *n* (%)**	
** Female**	56 (37)
** Male**	97 (63)
**Age, years**	67 (IQR 58–76)
** Marital status, n (%)**	
** Married**	88 (58)
** Divorced**	15 (9.8)
** Single**	13 (8.5)
** Widowed**	18 (12)
** Unknown**	19 (12)
**Palliative Performance Scale (PPS), *n* (%)**	
** ≤50%**	65 (42)
** ≥60%**	88 (58)
**Body mass index (BMI), kg/m^2^**	24.45 (IQR 21.8–27.0)
** Missing**	1
**BMI category, *n* (%)**	
** Normal weight**	75 (49)
** Overweight/obese**	65 (43)
** Underweight**	12 (7.9)
**Charlson Comorbidity Index (CCI)**	9 (IQR 6–16)
**Cancer type, *n* (%)**	
** Genitourinary**	43 (28)
** Lung**	35 (23)
** Breast**	26 (17)
** Gastrointestinal**	19 (12)
** Bone and soft tissue**	13 (8.5)
** Head and neck**	5 (3.3)
** Hematologic**	5 (3.3)
** Melanoma**	5 (3.3)
** Gynecological**	2 (1.3)
**RT fractionation, *n* (%)**	
** Hypofractionated (10–12 fractions)**	123 (80)
** Ultrahypofractionated (5 × 4 Gy)**	29 (19)
**RT completion**	139 (91)
**Systemic therapy, *n* (%)**	
** Chemotherapy**	64 (42)
** Immunotherapy**	22 (14)
** Targeted therapy**	17 (11)
** Endocrine therapy**	3 (2.0)
** Bone-directed therapy**	8 (5.2)
** Unknown**	39 (25)
**Discharge destination, *n* (%)**	
** Home with specialized outpatient palliative care**	53 (35)
** Inpatient palliative care**	22 (14)
** Rehabilitation**	8 (5.2)
** Acute care/inpatient hospital**	4 (2.6)
** Death**	3 (2.0)
** Other/unknown**	63 (41)

Data are presented as *n* (%) or median (interquartile range, IQR).

**Table 2 cancers-18-00061-t002:** Univariate and multivariable Cox proportional hazards regression analyses for overall survival. The table presents hazard ratios (HR) with 95% confidence intervals (CI) for each variable. Univariate Cox models were fitted for each covariate separately; all covariates were then included simultaneously in the multivariable model.

Univariate and Multivariable Cox Proportional Hazards Models							
Characteristic	Univariate Cox				Multivariable Cox		
	N	HR	95% CI	*p*-Value	HR	95% CI	*p*-Value
Sex	126						
female		—	—		—	—	
male		1.34	0.92, 1.94	0.13	1.61	1.06, 2.46	0.027
Age	126	0.99	0.97, 1.00	0.11	0.98	0.96, 1.00	0.050
Marital Status	126						
other		—	—		—	—	
married		0.96	0.67, 1.37	0.8	1.15	0.77, 1.72	0.5
BMI Category	126						
Normal weight		—	—		—	—	
Underweight		1.35	0.71, 2.57	0.4	1.27	0.66, 2.47	0.5
Overweight/Obese		0.89	0.62, 1.30	0.6	0.89	0.59, 1.34	0.6
PPS	126						
<60		—	—		—	—	
≥60		0.75	0.52, 1.08	0.12	0.62	0.41, 0.93	0.021
CCI	126						
≤10		—	—		—	—	
≥11		0.98	0.65, 1.47	>0.9	1.27	0.78, 2.08	0.3
RT Completion	126						
no		—	—		—	—	
yes		0.08	0.04, 0.16	<0.001	0.06	0.03, 0.12	<0.001

Abbreviations: CI = Confidence Interval, HR = Hazard Ratio.

**Table 3 cancers-18-00061-t003:** Overall survival events stratified by cancer subtype.

Cancer Subtype	N	Events (n)	Event Rate (%)
Genitourinary	43	35	81.4
Lung	34	30	88.2
Breast	25	19	76.0
Gastrointestinal	19	18	94.7
Bone and soft tissue	13	11	84.6
Head and Neck	5	5	100
Hematologic	5	2	40.0
Melanoma	5	4	80.0
Gynecological	2	2	100

**Table 4 cancers-18-00061-t004:** Logistic regression analysis for discharge destination. Univariate and multivariable logistic regression models assessing factors associated with discharge to home versus other destinations after palliative RT. Results are expressed as odds ratios (OR) with corresponding 95% CI.

Characteristic	Univariate Logistic Regression				Multivariable Logistic Regression		
	N	OR	95% CI	*p*-Value	OR	95% CI	*p*-Value
Sex	153						
female		—	—		—	—	
male		1.05	0.53, 2.12	0.9	0.93	0.45, 1.93	0.8
Age	153	1.02	0.99, 1.05	0.2	1.03	1.00, 1.06	0.11
Marital Status	153						
other		—	—		—	—	
married		1.52	0.77, 3.06	0.2	1.49	0.74, 3.09	0.3
PPS	153						
<60		—	—		—	—	
≥60		1.06	0.54, 2.10	0.9	1.11	0.54, 2.31	0.8
BMI	152						
normal		—	—		—	—	
non-normal		0.81	0.41, 1.57	0.5	0.77	0.39, 1.53	0.5

Abbreviations: CI = Confidence Interval, OR = Odds Ratio.

**Table 5 cancers-18-00061-t005:** Stratified Cox proportional hazards regression model for overall survival. The model was stratified by the PPS (<60 vs. ≥60) to account for violation of the proportional hazards assumption in the initial model. The table shows hazard ratios (HR) and 95% confidence intervals (CI) for all covariates included in the multivariable model. Model concordance was 0.68.

Variable	HR	95% CI	*p*-Value
Sex			
female	—	—	
male	1.86	1.20, 2.88	0.006
Age (years)	0.98	0.96, 1.00	0.015
Marital status			
other	—	—	
married	0.91	0.61, 1.37	0.7
BMI Category			
Normal weight	—	—	
Underweight	1.30	0.67, 2.51	0.4
Overweight/Obese	0.95	0.62, 1.44	0.8
Charlson Comorbidity Index			
≤10	—	—	
≥11	1.18	0.72, 1.93	0.5
RT Completion			
no	—	—	
yes	0.04	0.02, 0.10	<0.001

Abbreviations: CI = Confidence Interval, HR = Hazard Ratio.

## Data Availability

All data generated and analyzed during this study are available from the corresponding author upon reasonable request.

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
