# Peer review of "Palliative Performance Scale Predicts Survival in Patients with Bone Metastasis Undergoing Radiotherapy"

_cancers, 2025, doi:10.3390/cancers18010061_

Round 1
Reviewer 1 Report
Comments and Suggestions for Authors
From a biostats and clinical epidemiology point of view, here are some suggestions for the Authors:
- continuous covariates have to be reported only as median/IQR
- add the median follow-up for the whole cohort
- add the number of events, total and stratified by cancer subtype
- restricted mean survival time, explain the reader why you added this (correct!) info!
- what about to treat PPS as a time-dep covariate in a Mantel-Byar model? in such way, you will be able to estimate the PPS contribute too, that is impossible stratifying the cohort by PPS
Author Response
|
Reviewer comment |
Authors’ response |
|
Continuous covariates have to be reported only as median/IQR. |
Thank you for this suggestion. All continuous covariates (e.g., age, BMI, Charlson Comorbidity Index) are now consistently reported as median and interquartile range (IQR) throughout the manuscript, including the baseline characteristics and Results section. Mean ± standard deviation values were removed accordingly. |
|
Add the median follow-up for the whole cohort. |
We agree that reporting follow-up time is important. Given the palliative intent of treatment and the absence of structured follow-up after radiotherapy, all patients experienced the event of interest (death). Therefore, the median follow-up time corresponds to the median overall survival and is now explicitly reported in the Results section for clarity. |
|
Add the number of events, total and stratified by cancer subtype. |
This information has now been added. The total number of overall survival events as well as event numbers and rates stratified by cancer subtype are summarized in a new table (Table 5), providing a descriptive overview of outcome heterogeneity across tumor entities. |
|
Restricted mean survival time: explain the reader why you added this (correct!) info! |
We thank the reviewer for highlighting this point. A detailed explanation has been added to the Statistical Analyses section. Restricted mean survival time (RMST) was included as an absolute and clinically interpretable measure of average survival time that does not rely on the proportional hazards assumption and is particularly suitable for palliative cohorts with short and heterogeneous survival and limited follow-up. |
|
What about treating PPS as a time-dependent covariate in a Mantel–Byar model? In such way, you will be able to estimate the PPS contribution too, that is impossible stratifying the cohort by PPS. |
We appreciate this thoughtful methodological suggestion. A Mantel–Byar or other time-dependent approach requires longitudinal PPS assessments with clearly defined change times. In our dataset, PPS was assessed only at baseline prior to radiotherapy, and no repeated measurements during follow-up were available. Consequently, modeling PPS as a time-dependent covariate would not be methodologically justified. Instead, to appropriately address the violation of the proportional hazards assumption for PPS, we performed a stratified Cox regression as a sensitivity analysis. This approach allows valid estimation of the effects of other covariates while acknowledging non-proportionality of PPS. This rationale is now explicitly stated in the Methods and Sensitivity Analysis sections. |
We believe that these revisions have substantially improved the methodological transparency and robustness of the manuscript and we thank the reviewer for these valuable suggestions.
Reviewer 2 Report
Comments and Suggestions for Authors
First of all, I would like to congratulate the authors for their thorough and well conducted research. Overall, the study is significant in that it explores prognostication in a real-world palliative RT setting, which can help the practitioners in treatment planning and patient counseling. The manuscript is clear and well-organized, with a logical flow from background, methods, results, to discussion.
The use of a real-world, single-center cohort of palliative RT patients is a strength, as it reflects the diverse patient population encountered in routine practice. The key results are clearly presented and have clinical impact.
The study’s methodological rigor is sound: a retrospective cohort of adequate size (n=153) is analysed with appropriate statistical methods.
Suggestions for improvement:
Secondary Outcome (Discharge Destination): The inclusion of logistic regression analyzing discharge to home versus other destinations is a good addition, underlining the interest in patients’ post-treatment care needs. The result was that no factors (including PPS) significantly predicted discharge disposition.
It might be worth commenting briefly (in the Discussion section) on this negative finding. A short reflection on why no predictors emerged, or whether this outcome might require a larger sample to detect associations (41% of the patients had other or unspecified discharge destinations), would show the authors have thoughtfully considered this aspect. For instance, the authors could speculate that discharge destination is likely influenced by factors not captured in this study (such as availability of caregivers, social support, or rapid changes in condition post-RT). Currently, the discussion focuses almost entirely on survival, so one or two sentences about the discharge outcome (even if just noting that it did not correlate with PPS or other factors) would help gathering a better point of view on the results of the chosen secondary outcome.
I also noticed an inconsistency throughout the manuscript regarding the threshold selected for the PPS. Please clarify the rationale for choosing a cutoff value of 70 in the Kaplan–Meier analysis and a cutoff value of 60 in the Cox regression model.
Author Response
|
Reviewer comment |
Authors’ response |
|
Secondary Outcome (Discharge Destination): The inclusion of logistic regression analyzing discharge to home versus other destinations is a good addition. It might be worth commenting briefly on this negative finding in the Discussion section. |
We thank the reviewer for this insightful comment. We have now added a dedicated paragraph to the Discussion section addressing the negative finding of the discharge destination analysis. Specifically, we discuss that discharge disposition after palliative radiotherapy may be influenced by factors not captured in this study, such as availability of informal caregivers, social support, organizational aspects of post-acute care, and rapid changes in clinical condition after treatment. We also note that the relatively high proportion of patients discharged to other or unspecified destinations (41%) may have limited the statistical power to detect significant associations. This addition provides a more balanced interpretation of the secondary outcome beyond survival. |
|
Inconsistency regarding the threshold selected for the PPS (70% in Kaplan–Meier analysis vs. 60% in Cox regression model). |
We appreciate the reviewer for highlighting this inconsistency. To ensure methodological coherence and improve clarity, all analyses were harmonized to use a uniform PPS cutoff of 60%. This threshold is clinically established in palliative care and distinguishes patients with preserved ambulatory function from those with marked functional impairment. The Kaplan–Meier analysis was updated accordingly, and the rationale for selecting this cutoff is now explicitly stated in the Methods section, ensuring consistency across all analyses. |
Round 2
Reviewer 1 Report
Comments and Suggestions for Authors
The authors were able to properly solve all previous concerns, congrats and merry Xmas!